# Physiological Benefits of Dietary Lysophospholipid Supplementation in a Marine Fish Model: Deep Analyses of Modes of Action

**DOI:** 10.3390/ani13081381

**Published:** 2023-04-18

**Authors:** Antoni Ibarz, Ignasi Sanahuja, Waldo G. Nuez-Ortín, Laura Martínez-Rubio, Laura Fernández-Alacid

**Affiliations:** 1Department of Cell Biology, Physiology and Immunology, Faculty of Biology, University of Barcelona, Avda. Diagonal 643, 08028 Barcelona, Spain; 2IRTA, Centre de La Ràpita, Aquaculture Program, 43540 La Ràpita, Spain; 3Adisseo, Polígono Industrial, Valle del Cinca, 8A, 22300 Barbastro, Spain; 4Mowi Feed AS, Sandviksbodene 77AB, 5035 Bergen, Norway

**Keywords:** Atlantic salmon, dietary emulsifiers, interactome, lipid metabolism, post-translational modifications

## Abstract

**Simple Summary:**

While lysophospholipid supplementation in animal feed mainly aims at improving fat emulsification, additional metabolic effects in fish have not been explored. The present study aimed to gain understanding of the mechanisms underlying the growth-promoting effect of lysophospholipid supplementation using a commercial product (AQUALYSO^®^, Adisseo, Barbastro, Spain). Atlantic salmon was selected as the model species with the intestine and liver as target tissues. The resulting tissue interactomes revealed the putative mode of action of LPLs in salmon nutrition. In summary, the biological processes stimulated by the LPL-diet suggest a more robust digestive capacity together with better nutrient processing in the liver, favoring the conversion of nutrients into weight gain and showing a less-reactive intestine and liver condition. These nutraceutical effects would improve the ability of fish or other organisms to deal with production conditions or with modifications in the diet.

**Abstract:**

Given the hydrophilic structure of lysophospholipids (LPLs), their dietary inclusion translates into a better emulsifying capacity of the dietary components. The present study aimed to understand the mechanisms underlying the growth-promoting effect of LPL supplementation by undertaking deep analyses of the proximal intestine and liver interactomes. The Atlantic salmon (*Salmo salar*) was selected as the main aquaculture species model. The animals were divided into two groups: one was fed a control diet (C-diet) and the other a feed (LPL-diet) supplemented with an LPL-based digestive enhancer (0.1% AQUALYSO^®^, Adisseo). The LPL-diet had a positive effect on the fish by increasing the final weight by 5% and reducing total serum lipids, mainly due to a decrease in the plasma phospholipid (*p* < 0.05). In the intestine, the upregulated interactome suggests a more robust digestive capacity, improving vesicle-trafficking-related proteins, complex sugar hydrolysis, and lipid metabolism. In the liver, the LPL-diet promotes better nutrients, increasing several metabolic pathways. The downregulation of the responses to stress and stimuli could be related to a reduced proinflammatory state. This study on the benefits and modes of action of dietary LPLs opens a new window into fish nutrition and could be extended to other productive species.

## 1. Introduction

Current trends in aquafeed formulation not only guide the optimization of nutritional inputs and ingredient selection as affordably as possible but also promote the optimal functioning of the digestive system and whole metabolism in fish. One current strategy is increasing the lipid utilization coefficient in order to reduce lipid content in fish feed, decrease metabolic pressure, and increase the proportion of energy provided by lipids (protein sparing) [1]. Digestibility enhancers, such as lysophospholipids (LPLs), that maximize the efficiency of digestive and metabolic processes can be used as alternative ingredients in the formulation of cost-effective and high-quality feeds.

The potential of LPLs, mainly lysolecithins, has been investigated previously in broilers [2,3], piglets [4], ruminants [5], and even some fish species [6,7]. Unlike phospholipids (lecithins), LPLs (lysolecithins) contain only one fatty acid tail, making them more hydrophilic [8]. This translates into a better emulsifying capacity, meaning a better capacity to disperse lipids and form smaller and an increased number of micelles, consequently leading to better absorption of lipid components. Lecithins are a by-product of the processing of vegetable oils, with phospholipids being the main constituents. Lysolecithins are produced by an enzymatic conversion comprising a hydrolytic reaction via the phospholipase A1 or A2. Lysolecithins are therefore a mixture of phospholipids and lysophospholipids, which differ in their phosphatidyl substituent and fatty acid pattern [8,9,10]. While many studies have demonstrated the beneficial effects of dietary phospholipids on fat digestibility in fish [11], like in mammals and birds, the effect of LPLs on growth performance and fat utilization has been shown only recently in carp (*Carassius auratus*; [6]), turbot (*Scophthalmus maximus*; [12]), rainbow trout (*Oncorhynchus mykiss*; [13]), and channel catfish (*Ictalurus punctatus*; [14]). Although these limited studies provide some evidence of improvements in growth performance, limited knowledge exists mainly on whether the LPL is effective and on what the effective LPL dose is. Moreover, the physiological mechanisms underlying the modes of action or additional metabolic effects of LPLs have not been explored yet.

Fish farming of the Atlantic salmon has become a large industry worldwide. As one of the most important species in aquaculture, it is the perfect candidate to study the benefits of this type of emulsifier. The Atlantic salmon represents 32.5% of global marine finfish aquaculture production and is the fourth most economically important farmed species. Current worldwide production of farmed Atlantic salmon exceeds 1,000,000 tonnes, where farmed Atlantic salmon constitute over 90% of the farmed salmon market and over 50% of the total global salmon market (FAO, 2022; https://www.fao.org, accessed on 20 January 2023). Moreover, salmon flesh coloration, which is a key commercial trait of wild and farmed salmon alike [15], can be improved by the lipid-associated absorption of carotenoids [16]. Therefore, the present study in juvenile salmon examined the physiological effects of a commercial LPL product on the main function of the intestine and liver, as well as on fish growth and plasma lipid fractions. We undertook proteomics, involving the identification, localization, and quantification of differentially expressed proteins (DEPs), and also analyzed protein modifications and protein–protein networks [17]. The resulting tissue interactomes revealed the putative mode of action of LPLs in salmon nutrition. Thus, the results of this study could be beneficial in better understanding the effects of LPLs on growth performance, lipid utilization, enterocyte function, and liver metabolism, as well as in applying LPLs in fish feeds and global aquaculture.

## 2. Materials and Methods

### 2.1. Feeding Trial

The feeding trial was carried out in 8 indoor tanks (surface area of 2 m^2^ and a water volume of approximately 1.2 m^3^) at Nofima’s Research Station for Sustainable Aquaculture in Sunndalsøra, Norway. One week before the start of the feeding trial, the tanks were stocked with 35 fish, with an average weight of 160 g, and supplied with seawater (salinity approximately 33 ppt at an ambient temperature). The experimental feeds were then administered to two triplicate groups of Atlantic salmon in a 12-week experiment: the control group and the LPL-supplemented group (LPL-group). Feeds were produced as 5 mm pellets according to fish size from a formulated feed (dry matter: 94%; crude protein: 44.5%; fat: 28.8%; and gross energy: 24.2 MJ/kg), without (control group) or with (LPL-group) 0.1% of an LPL-based additive in liquid form (AQUALYSO^®^, Adisseo). The diets were based on a common basal formulation for salmon with 20% of fish meal and the LPLs were added to the mixer before extrusion replacing 0.1% of the rapeseed oil.

The water flow was equally adjusted in all the tanks (24 L·min^−1^) and kept at a level sufficient to maintain oxygen concentrations at >80% saturation. The fish were kept under continuous light (L:D 24:0). They were fed by automatic disc feeders, which delivered one meal every 3 h and 8 meals per day. An overfeeding by 20% was undertaken to allow maximum feed intake. All the feed spill was collected from the water outlet and the feed intake was quantified as the difference between the delivered and wasted feed, after correcting for dry matter content and feed spill recovery. The tanks were controlled by programmable logic controllers (PLCs), which enabled the continuous logging of research data, including the pump status, water flow, temperature, oxygen, pH, and the oxidation/reduction potential, and were connected to an autoanalyzer to measure levels of ammonia, nitrite, nitrate, and total inorganic carbon.

At the end of the trial, ten fish per condition were randomly sampled to obtain serum and tissues, with the remaining animals used to determine growth performance. The fish were euthanized with an overdose (400 mg·L^−^^1^) of metacaine (tricaine methanesulfonate, MS 222, Argent Chemical Laboratories, Redmont, WA, USA), before their weight and length were measured. Blood samples were taken by caudal puncture, utilizing serum tubes (BD Vacutainer^®^). The tubes, stored at 4 °C, were centrifuged at 5000× *g* for 10 min. The serum was aspirated and stored at −80 °C. Immediately after sacrifice, the liver and proximal intestine of the fish were removed and kept at −80 °C until analysis. Fish growth was evaluated by means of the following indices: Fulton’s condition factor (K) = (BW_f_/SL_f_^3^) × 100; specific growth rate in BW (SGR_BW_, %) = ((ln BW_f_ − ln BW_i_) × 100)/time (d), where BW_f_ and BW_i_ correspond to final and initial body weight (BW) and SL_f_ corresponds to final standard length (SL), respectively. Feed utilization was evaluated by the following formula: feed conversion ratio (FCR) = feed intake (g)/increase of fish biomass (g). Thermal growth coefficient (TGC) was calculated as follows: TGC = 1000 × (BW_f_^1/3^ − BW_i_^1/3^) (T × t), where T is temperature in °C and t is time in days. The research facilities are licensed under Norwegian law to perform experimental work on fish. The regulation in question is FOR-2015-06-18-761 «Forskrift om bruk av dyr i forsøk» and is a national adaptation of (and in compliance with) the “Directive 2010/63/EU of the European Parliament an of the Council on the protection of animals used for scientific purposes”.

### 2.2. Blood Biomarkers

Serum samples were analyzed for different known biomarkers: total protein, glucose, lactate dehydrogenase (LDH), and the lipid fractions of triacylglycerides (TAG), phospholipids (PL), and total cholesterol (TC), as well as the sum of circulating lipids (SL). Plasma protein was measured using the method of Bradford and Williams (1976) [18], with bovine serum albumin used as a standard. Glucose and lipid fractions were measured in triplicate in pre-diluted plasma samples, when necessary, using an endpoint colorimetric assay and each internal standard of the respective kits from Spinreact (Girona, Spain). LDH activity was measured by determining the rate of decrease in the concentration of NADH, measured photometrically, according to the specification of the kit. All measurements were obtained with a microplate reader (Infinite 200 PRO spectrophotometer, Tecan, Spain). Plasma concentrations are expressed in mg/mL (protein), mg/dL (glucose and lipid fractions), or U/mL (LDH).

### 2.3. Sample Preparation for Proteomics

Tissue samples (around 50 mg) were individually homogenized in Eppendorf tubes containing lysis buffer (7 M urea, 2 M thiourea, 2% *w*/*v* CHAPS, and 30 mM Tris-HCL, pH 7.4) and a protease inhibitor cocktail powder (Sigma-Aldrich, St. Louis, MS, USA) at a ratio of 1:4, *w*/*v*, at 4 °C, as described for fish tissues in Ibarz et al. (2010) [19]. The homogenate was rested for 30 min at 4 °C, being vortexed every 5 min to improve protein extraction. Then, the samples were centrifuged (20,000× *g* for 15 min at 4 °C) and the soluble fraction was aliquoted. The protein concentration of homogenized tissues was determined using the Bradford assay. Five pools from two samples for each condition and tissue were prepared and adjusted with lysis buffer to achieve a concentration of 1 μg·μL^−1^. The five resulting pools per tissue were stored at −80 °C until shotgun analysis.

### 2.4. Proteomics Data Acquisition

The shotgun analyses were conducted with the i3S Proteomics Platform (Porto, Portugal). Briefly, 100 μg of protein from each sample was processed for proteomic analysis, following the solid-phase-enhanced sample preparation (SP3) protocol. Enzymatic digestion was performed with trypsin/LysC (2 µg) overnight at 37 °C at 1000 rpm [20]. Protein identification and quantification were performed with nano LC-MS/MS, which consisted of an UltiMate 3000 liquid chromatography system coupled to a Q Exactive Hybrid Quadrupole-Orbitrap mass spectrometer (Thermo Scientific, Bremen, Germany). Five hundred nanograms of peptides of each sample were loaded into a trap cartridge (Acclaim PepMap C18 100 Å, 5 mm × 300 µm, i.d., 160454; Thermo Scientific, Bremen, Germany) in a mobile phase of 2% ACN and 0.1% FA at 10 µL/min. After 3 min of loading, the trap column was switched in-line to an EASY-Spray column (with an inner diameter of 50 cm × 75 μm; ES803, PepMap RSLC, C18, 2 μm; Thermo Scientific, Bremen, Germany) at 250 nL/min. Separation was achieved by mixing A (0.1% FA) and B (80% ACN and 0.1% FA) at the following gradient: 5 min (2.5% B to 10% B), 120 min (10% B to 30% B), 20 min (30% B to 50% B), 5 min (50% B to 99% B), and 10 min (hold at 99% B). Subsequently, the column was equilibrated with 2.5% B for 17 min. Data acquisition was performed with the Xcalibur 4.0 and Tune 2.9 software (Thermo Scientific, Bremen, Germany).

The mass spectrometer was operated in the data-dependent (dd) positive acquisition mode, alternating between a full scan (*m*/*z* 380–1580) and the subsequent HCD MS/MS of the 10 most intense peaks from a full scan (normalized collision energy of 27%). The ESI spray voltage was 1.9 kV. The global settings were as follows: use lock mass best (*m*/*z* 445.12003), lock mass injection Full MS, and chromatography peak width (FWHM) of 15 s. The full scan settings were as follows: 70 k resolution (*m*/*z* 200), an AGC target of 3 × 106, and a maximum injection time of 120 ms, with the dd settings of a minimum AGC target of 8 × 103 and an intensity threshold of 7.3 × 104, while the charge exclusion was unassigned, 1, 8, >8, peptide match preferred, and with isotopes excluded and a dynamic exclusion of 45 s. The MS2 settings were as follows: 1 microscan, a resolution of 35 k (*m*/*z* 200), an AGC target of 2 × 105, a maximum injection time of 110 ms, an isolation window of 2.0 *m*/*z*, an isolation offset of 0.0 *m*/*z*, and the use of dynamic first mass and a spectrum data type profile.

### 2.5. Data Analysis

The raw data were processed using the Proteome Discoverer 2.5.0.400 software (Thermo Scientific, Bremen, Germany). Protein identification was performed with the data available in the UniProt database for salmon, as well as with MaxQuant (version 1.6.2.6 for common contaminants, Max Planck Institute of Biochemistry, Munich, Germany). The tolerance levels for matching to the database was 6 ppm for MS and 20 ppm for MS/MS. Trypsin was used as the digestion enzyme, and two missed cleavages were allowed. The carbamidomethylation of cysteine residues was set as a fixed modification, while N-terminal acetylation and methionine oxidation were allowed as variable modifications. The “match between runs” feature of MaxQuant, which enables identification transfer between samples based on accurate mass and retention time [21], was applied with a match time window of one minute and an alignment time window of 20 min. All identifications were filtered to achieve a protein false discovery rate (FDR) of 1% and further filtering was applied to include at least one unique peptide and at least two peptides in total.

### 2.6. Protein Functional Enrichment and Network Analysis

To determine the differentially expressed proteins (DEPs) between the LPL-diet, the control diet, and the control cases, the following filters were considered: (1) the minimum number of samples that a protein had to be detected in to be used was set to 60% (3 of 5) per experimental group; (2) at least two unique peptides were used and the *p*-value was adjusted using the Benjamini–Hochberg correction for the FDR set to ≤0.05; and (3) the difference between the LPL and control groups had to show a fold change of ≥1.50 for the selection of the upregulated and downregulated proteins. Volcano plots were obtained with the Proteome Discoverer software after applying the above-described filters.

Protein functional enrichment analysis was performed using the Search Tool for the Retrieval of Interacting Genes/Proteins (STRING) public repository, version 10.0 (https://string-db.org, accessed on 30 November 2022) [22]. Protein–protein interaction (PPI) networks for the DEPs were obtained with a high-confidence interaction score (0.9). The mechanisms of response involving the DEPs were obtained from a comparative analysis using *Homo sapiens* as a reference organism in order to extract the maximum information currently available. Thus, an ortholog *H. sapiens* Entrez Gene ID was assigned based on sequence homology. The UniProt (Consortium, T.U., UniProt, 2019) and GeneCards databases [23] were used to match the gene acronym tag between the two species. The detailed list of human orthologs is shown in Appendix A. Gene ontology (GO) enrichment analysis (biological processes, Reactome pathways, and annotated keywords) was also performed for the DEPs using STRING, performing a Fisher’s exact test followed by a correction for multiple testing [24]. A *p*-value < 0.05 was considered significant. The GO terms obtained were then identified in the ancestor GO chart using the QuickGO web-based tool (https://www.ebi.ac.uk/QuickGO/, accessed on 30 November 2022) [25]. The GO chart for each GO obtained from the enrichment analysis was then mapped onto a single chart to identify the less redundant GO terms.

### 2.7. Statistical Analysis

Data on growth performance and serum biomarkers were compared by Student’s *t*-test using a commercial software (PASW version 21.0, SPSS Inc., Chicago, IL, USA). The data were checked for normality and homoscedasticity prior to analysis. The STRING database was used to obtain direct protein–protein interactions (PPI), the interactome, using the STRING program v10.5 [22]. The selected stat indicators were “clustering coefficients” and “PPI-enrichment *p*-value”, which correspond to a measure of how connected the nodes in the network are, and “count in gene set”, which indicates the number of proteins included and their false discovery rate. The enrichment tests from the STRING software are conducted for a variety of classification systems (Gene Ontology, KEGG, Pfam, and InterPro) and employ a Fisher’s exact test followed by a correction for multiple testing [24].

## 3. Results

### 3.1. Growth Performance and Plasma Biomarkers

The LPL-enriched diet had a positive effect on fish by increasing the final weight by 5% when compared to the fish on the control diet (*p* < 0.05) after the 3-month feeding period. However, there was no effect on fish length, resulting in an increased condition factor (k, Table 1). The LPL-diet also increased liver size by over 10% (*p* < 0.05). There were no significant differences in the rest of performance parameters, such as the specific growth rate (SGR), the thermal growth coefficient (TGC), and the feed conversion rate (FCR) (data in Table 1).

The effects of the LPL-diet on plasma biomarkers such as protein, glucose, and specific lipid fractions (triglycerides, phospholipids, and total cholesterol) are summarized in Table 2. Moreover, the activity of LDH, a secondary stress biomarker, was measured. Whereas plasma protein levels were not affected by dietary LPLs, total circulating plasma lipid levels were significantly reduced, mainly due to a lower plasma PL content (736 ± 35 and 666 ± 15 mg/dL for the C-diet and LPL-diet, respectively, *p* < 0.05). With regards to LDH activity in the plasma, fish on the LPL-diet showed significantly reduced values (by 30%) compared to those on the C-diet.

### 3.2. Differentially Expressed Proteins

Quantitative analysis from the shotgun proteomics of the salmon intestine and liver identified a total of 4984 and 4850 proteins, respectively, using the Atlantic salmon database. Figure 1 shows the differentially expressed proteins, or DEPs, and the resulting Volcano plot. In the intestine, 187 DEPs were detected, of which 61 were upregulated (Appendix A) and 126 downregulated (Appendix A), with only 3 proteins being uncharacterized and discarded from further analyses. In the liver, 194 DEPs were detected, of which 114 were upregulated (Appendix A) and 80 downregulated (Appendix A), with only 10 proteins that could not be characterized and were subsequently discarded from further analyses. Details on protein identification are supplied in each corresponding Appendix A, providing the protein identity, their accession number, the corresponding gene number, the UniProtKB symbol, the identified peptides (peptide score, coverage, isoforms, unique peptides, and the expected MWs and pI), the abundance ratio (the C-diet vs. the LPL-diet), and the *p*-value.

### 3.3. Effects on Enterocyte Function

Figure 2 presents the obtained DEP networks (interactomes or proteinteractomes) in the intestine for the total DEPs, while Figure 3 shows them for the upregulated DEPs (Figure 3A) and downregulated DEPs (Figure 3B). The LPL-diet had a clear impact on enterocyte organization, as evidenced by the appearance of the GO term “cellular component organization” (GO:0016043; 70 DEPs included), particularly on the directed movement of substances into, out of, or within a cell (“transport”, GO:0006810; 57 DEPs included). Interestingly, among the Reactome pathways, the LPL-diet modified “metabolism” (HSA-1430728, 45 DEPs included), particularly the “metabolism of lipids” (HSA-556833, 17 DEPs included). From the functional enrichments in the PPI-enrichment network, two interesting annotated keywords were revealed, phosphoprotein (KW-0597) and acetylation (KW-007), both related to post-translational protein modifications, which included 94 and 61 DEPs, respectively, indicating a possible involvement of LPLs in these pathways.

Within the cellular reorganization caused by dietary LPLs, the upregulated DEPs in the intestine built an interactome strongly associated with the “intracellular organelle”, significantly clustering 50 of the 57 upregulated DEPs (Figure 3A). Interestingly, deeper analyses of these 50 proteins related to enterocyte organelle function indicated that the following specific pathways were improved by dietary LPLs (Figure 4 and Appendix A): vesicle trafficking, mucus formation, and cellular metabolism. Twelve DEPs were differentially clustered (PPI-enrichment *p*-value = 0.8 × 10^−3^) into vesicle-trafficking-related processes (Figure 4A), such as endosome and exosome vesicles (AAPL1 and CORO1A, and ARL3 and WDR44, respectively), Golgi vesiculation (CPAb1 and STRN4), and vesicle intracellular movement. The activity of intestinal mucous cells seemed to be also improved by LPLs, with the upregulation of five DEPs associated with mucin exudation: two types of keratins (KRT13 and KRT15), two mucins (MUC2 and MUC5), and an enzyme (GALNT12). A broad group of upregulated DEPs in the intestine of LPL-fed animals was associated with cell metabolism (also differentially clustered, Figure 4B), such as the hydrolysis of complex sugars, glycolysis, lipid metabolism, and proteasome activity (see specific DEPs in Appendix A). Finally, other upregulated DEPs occurred in several linked pathways related to protein synthesis, such as mRNA maturation, protein translation, protein location, and post-translational modifications (PTMs) of newly formed proteins.

Among the 119 downregulated DEPs obtained in the intestine, only the proteins (54) clustering in “cellular component organization” (Figure 3B, PPI-enrichment *p*-value = 6.9 × 10^−4^) were grouped in a subsequent interactome with relevant biological processes affected (Figure 5). For instance, there were 19 proteins clustered in the “response to stress” (GO:0006950), 7 in the “viral process” (GO:0016032), and 4 in “type I interferon signaling pathway” (GO:0060337), some of them belonging to two of these clusters and one to all. The rest of the DEPs downregulated in the intestine by dietary LPLs did not cluster in any further subgroup using the information available.

### 3.4. Effects on Hepatocyte Function

Figure 6 shows the interactome of the DEPs (174 proteins with a PPI-enrichment *p*-value < 1.0 × 10^−16^) in the livers of animals fed the LPL-diet, including the main functional enrichment clusters. The most relevant biological process was “metabolic process” (GO:0008152), clustering 110 DEPs. Among the Reactome pathways with significant grouping were “metabolism” (HSA-1430728, with 56 DEPs), including “lipid metabolism” (HSA-556833, with 18 DEPs) and the “innate immune system” (HSA-168249, with 21 DEPs). Similar to that observed in the intestine, several DEPs were susceptible to post-translational modifications: 103 DEPs could be phosphorylated (KW-0597) and 72 DEPs could be acetylated (KW-007), indicating a possible involvement of dietary LPLs in these pathways.

The network of the upregulated DEPs in the liver is shown in Figure 7, including 101 proteins with a high PPI-enrichment *p*-value of 1.1 × 10^−16^. With regards to the biological processes affected, most of the DEPs belonged to “metabolic process” (69 DEPs) or “transport” (42 DEPs) (Figure 7A). Interestingly, when analyzing the metabolic pathways that were upregulated in the liver by dietary LPLs, the Reactome pathways (Figure 7B) revealed that “lipid metabolism” (14 DEPs), “amino acid metabolism” (9 DEPs), and “carbohydrate metabolism” (6 DEPs) were targets, as well as the tricarboxylic acid cycle. Figure 8 presents the network of the 74 downregulated DEPs in the liver, with a weak PPI-enrichment *p*-value of 0.012. Functional enrichment analysis revealed that the main effect of the LPLs was related to the “response to stimulus” (GO:00508096, including 49 DEPs), matching the downregulation of the response to stress in the intestine.

### 3.5. Summary of Putative Modes of Action of Dietary LPLs

The functions of both the intestine and liver were modified in salmon supplemented with 0.1% of an LPL-based additive for 3 months. Figure 9 summarizes the modes of action of dietary LPLs, resulting from the analyses of the intestinal and liver interactomes and each functional enrichment, focusing on the most relevant biological processes, Reactome pathways, and post-translational modifications (PTMs). A clear improvement in enterocyte function mediated by dietary LPLs was evidenced via the stimulation of vesicle trafficking, complex sugar hydrolysis, and lipid metabolism, together with a putative increase in mucus production. Dietary LPLs also seemed to stimulate liver activity, enhancing the metabolism of lipids, carbohydrates, and amino acids, as well as increasing the energy obtained via the Krebs cycle. In parallel, both tissues showed reduced reactivity, with downregulations in the stress response (intestine) or in the response to stimuli (liver). Finally, a great number of the DEPs, upregulated and downregulated, were targets of PTMs, particularly phosphorylation and/or acetylation.

## 4. Discussion

In modern aquaculture, it is advisable to have certain flexibility in the use of alternative ingredients in the formulation of cost-effective and high-quality feeds. It is also necessary to find alternative sustainable sources of protein and lipids as well as alternative products that can improve nutrient absorption and help animals cope with productive conditions. Digestibility enhancers aim to maximize lipid absorption and utilization, and are categorized as emulsifiers [2,3,4,5,6,7]. A wide variety of products improving nutrient uptake and utilization have been investigated in livestock species. However, the feeding behavior, digestive physiology, and nutritional requirements of aquaculture organisms might strongly differ from those of livestock species [11,12,13,14]. The current work is the first study in aquaculture species analyzing the effects of dietary LPLs (from vegetable by-products) on the function of the intestine and liver in a leading piscine species, the Atlantic salmon. Our approach used protein interactome networks to elucidate putative modes of action.

Improvements in fish growth and/or feed utilization caused by dietary lysolecithin supplementation have been reported previously in common carp, crucian carp, catfish, and turbot [6,7,26,27], as well as terrestrial species. The results of the present study agree, showing a significant increase in body weight of 5% in salmon after 12 weeks of being fed an LPL-supplemented diet. The SGR also showed a tendency for higher values, whereas the FCR did not change. The mode of action of emulsifiers involves an increase in the active surface of lipids and the promotion of fatty acids to form micelles, which is crucial in lipid digestion and absorption [28], as well as for higher energy availability in animals. In parallel, a few studies have suggested that the level of dietary lysolecithin supplementation could decrease body lipid deposition in fish (Liu et al., 2019). However, there are some discrepancies regarding this effect depending on the fish species and on the different life stages and environmental conditions [7,8,9,10,11,12,13,14,15,16,17,18,19,20,21,22,23,24,25,26]. In the present study, the effect of supplementation significantly affected fish final weight and the condition factor, and numerically the SGR and TGC, while no numerical or significant effect was detected in the FCR. The current study was performed in a short time period considering the entire cycle of farming salmon. Further studies are suggested to better understand the potential growth performance effect of lysophospholipids through the farmed Atlantic Salmon life cycle.

Hematological parameters can directly indicate the physiological and pathological status as well as the health of fish [29]. We analyzed the levels of circulating lipid, including lipid fractions such as TAG, PL, and TC, under pre-feeding conditions (samples were obtained after overnight fasting). There are no data on the effects of dietary LPLs on circulating PL levels in the literature, but Liu et al. [27] did detect lower levels of circulating TAG and TC in LPL-fed channel catfish. Furthermore, we previously demonstrated in gilthead sea bream that diets with higher lipid levels corresponded to lower sums of lipids in the plasma after overnight fasting [30]. Thus, as the control diet and the LPL-diet used in the present study did not differ in lipid content (28.8%), the presence of LPLs would enhance lipid intake in the fish, meeting their daily needs. We also measured LDH activity in the plasma to detect any damage in the tissues [31]. Fish on the LPL-diet showed reduced LDH activity, which could be related to a better condition.

Proteomics is widely used to understand fish physiology, as this technique can show differential expression of identified proteins at various stages of fish development and under different conditions of feeding, stress, or disease, thereby providing a holistic overview of several functions in fish metabolism [32]. Here, using shotgun proteomics, we identified a high number of proteins (4850 and 4984 for the intestine and liver, respectively), like other studies using shotgun approaches [33,34,35] which were greater than those of other studies using 2D proteomics in fish tissues [19,36]. Next, we used GO enrichment analysis for the DEPs, similar to recent proteomic evaluations in fish [35,37], mainly searching for the biological processes and Reactome pathways affected by dietary LPLs. According to the functional enrichment analyses, the addition of LPLs to the diet affected enterocyte function in two principal ways: cellular organization (GO:0016043, “cellular component organization”, 70 DEPs), a process “that results in the assembly, arrangement of constituent parts, or disassembly of a cellular component” (GO strict definition), and cellular transport (GO:0006810, transport, 57 DEPs), which is “the directed movement of substances or cellular components (such as complexes and organelles) into, out of or within a cell, or between cells” (GO adapted definition). To date, there are no studies on how enterocyte functions are directly affected by dietary LPLs. For the first time in fish, we conducted deep analyses of the enterocyte interactome, with the upregulated DEPs revealing some specific protein–protein interactomes promoted by dietary LPLs. The vesicle trafficking protein network (12 DEPs with a PPI-enrichment *p*-value of 8.1 × 10^−4^) indicated that LPLs favor the intake and transport of macromolecules, which is consistent with the studies in mammals reporting that LPLs could modify the lipid bilayer of the membrane, altering membrane fluidity and the transmembrane permeability of nutrients, thus promoting nutrient digestibility [38,39]. We also present a novel finding that improved vesicle trafficking could be the basis for a better exudation of mucus and the maintenance of intestinal barrier integrity (e.g., the upregulation of some mucins, periplakin, and keratins). Regarding these findings on enterocyte functionality, further studies are necessary in approaching the role of the activity of digestive enzymes related to lipid digestion and absorption in fish fed with LPLs.

In terms of enterocyte metabolism, we detected 18 upregulated proteins, which clustered with a high PPI-enrichment *p*-value. As expected, lipid metabolism was improved by dietary LPLs, as suggested previously [6,7,27]. LPLs increase glucose uptake in rainbow trout (*Oncorhynchus mykiss*; [40]) as well as in broiler chickens [41]. Glucose is generally utilized as a source of energy and a metabolic intermediate, as indicated by some of the upregulated DEPs obtained in this study (e.g., the TREH, HEXA, and GLB1 proteins are associated with the hydrolysis of complex sugars in the lumen, while the ENO1, ENOPH1, and G6PD proteins are related to glycolysis). Interestingly, another group of upregulated proteins in enterocytes, although not directly linked, are associated with protein synthesis and turnover functions. However, no clear information is available on this subject regarding the LPLs. In medical studies in mammals, LPLs have been shown to elicit a wide range of biological effects, including cell proliferation, cellular signaling processes, calcium mobilization, metabolic activity, inflammatory and anti-inflammatory processes, and neuritogenesis (reviewed in [42]). Here, we observed a link between dietary LPLs and increased proteasomal activity (upregulation of the proteins GIMAP7, PSMA1, PSMB1, RFPL4A, and UBA1), mRNA maturation (SYF2, SRSF7, and SRSF2), and the folding and final expression and location of proteins. These results open a new window into the effects of LPLs beyond lipid metabolism, which should be further investigated by taking into account the capacity of LPLs to act as signaling molecules [43].

The liver is sensitive to the nutritional condition in fish. One of the expected effects of dietary LPLs in the liver is the prevention of the accumulation of abnormal fats [44]. For instance, the inclusion of lysolecithin in the diet of channel catfish led to a lower lipid content in the liver and increases in intestinal lipase activity [27], which were also observed in rainbow trout [13]. This might be because LPLs facilitate the transport of the lipids produced from lipoprotein synthesis from the liver to the adipose tissue. Accordingly, we observed that the main effect of dietary LPLs in the livers of LPL-fed salmon corresponded to the upregulation of the “lipid metabolic process” (GO:0006629, 20 DEPs, including the Reactome pathway of “metabolism of lipids” with 14 DEPs). Therefore, we demonstrate here for the first time a direct link between dietary LPLs and improved liver lipid metabolism. For instance, some of the proteins upregulated by dietary LPLs are relevant metabolic enzymes: long-chain fatty acid—CoA ligase 1 (ACSL1), which activates long-chain fatty acids for both the degradation via beta-oxidation and synthesis of cellular lipids [45]; cytochrome P450 1A1 (CYP1A1), which is involved in the oxidation of a variety of structurally unrelated compounds, including steroids, fatty acids, and xenobiotics [46]; fatty acid amide hydrolase 2 (FAAH2), which degrades bioactive fatty acid amides [47]; acyl-CoA desaturase (SCD), which plays an important role in regulating the expression of genes that are involved in regulating mitochondrial fatty acid oxidation [48]; ATP-binding cassette sub-family D member 3 (ABCD3), which is a probable transporter involved in the transport of branched-chain fatty acids and C27 bile acids into the peroxisome [49], the latter function being a crucial step in bile acid biosynthesis; and fatty-acid-binding protein (FABP2), which plays a role in the intracellular transport of long-chain fatty acids and their acyl-CoA esters [50]. FABP2 is probably involved in triglyceride-rich lipoprotein synthesis, favoring lipid release from the liver.

Alongside improved liver lipid metabolism, the LPL-diet seemed to enhance the function of whole hepatocytes, with the interactome of the upregulated DEPs including the following Reactome pathways: “metabolism of amino acids” (HSA-71291, clustering 9 DEPs significantly), “metabolism of carbohydrates” (HSA-556833, 6 DEPs), and the “citric acid cycle and respiratory electron transport” (HSA-1428517, including 5 DEPs). To our knowledge, there are currently no data on the relationship between dietary LPLs and these specific pathways in the liver in fish or in terrestrial animals. However, several studies have reported that LPLs act as potent hormone-like cellular mediators via the activation of cell-surface G-protein-coupled receptors (GPCRs) and as intracellular second messengers through peroxisome-proliferator-activated receptor gamma (PPAR-gamma) [42,51,52]. Thus, these unexpected upregulated modes of action of dietary LPLs should encourage new research in this field in fish and in terrestrial species where LPLs are used as emulsifiers, surfactants, and attractant additives. In the view of current results in intestine and liver functionality, the study of flesh quality and the applicability of this diet to obtain a high-quality product could be a next step with major interest of the salmon industry.

Regarding the downregulated DEPs in the intestine and liver of LPL-fed fish, they clustered mainly in biological processes related to tissue reactivity, such as the “response to stress” (GO:0006950, with 19 DEPs), which included 11 DEPs significantly grouped as “viral process” (7 DEPs) and “interferon signaling pathway” (4 DEPs) in the intestine, and the “response to stimulus” (including 49 of the 74 downregulated DEPs) in the liver. Supplemental bile acids, which have similar emulsifying properties to those of LPLs, have been shown to increase the activities of antioxidant enzymes and alleviate the damage to the antioxidant system caused by high fat levels in livestock [5]. In fish, Liu et al. [27] reported that the response to stimulus is involved in the upregulated response to handling and confinement stress in trout, while Raposo de Magalhaes et al. [53] also referred to this GO term in the analysis of the stress-responsive hepatic proteome in gilthead sea bream. In parallel, a recent study [54] in largemouth bass (*Micropterus salmoides*) demonstrated that the addition of LPLs increased the abundance of beneficial microbiota and decreased the abundance of harmful microbiota in the intestinal flora, which could also be related to the proposed less-reactive condition of fish receiving LPL supplementation. We attribute our novel results on the response to stress to a general reduced proinflammatory state in LPL-fed fish.

Finally, the study of changes in the proteinteractome caused by dietary LPLs revealed that most of the DEPs detected were putative targets of post-translational modification (PTM), mainly acetylation and phosphorylation. PTMs are reversible processes that can dynamically regulate the metabolic state of cells through regulating protein structure, activity, localization, or protein–protein interactions. PTMs can activate or inactivate catalytic functions or influence the biological activity of a protein [55]. There does not exist specific information on the direct relationship between LPLs and PTMs, but some evidence indicates a clear role of LPLs in cellular signaling. Diverse effects of LPLs on cell responses include mitogenesis, differentiation, cell migration, and cell viability (anti-apoptosis) (reviewed in [56]). As reversible protein phosphorylation and acetylation are major modifications regulating protein function, and lipid second messengers can modulate these cellular signaling pathways [56], the role of the dietary LPLs needs to be investigated, and further and deeper studies are necessary in this field.

## 5. Conclusions

In summary, the biological processes stimulated by the LPL-diet suggest a more robust digestive capacity together with better nutrient processing in the liver, favoring the conversion of nutrients into weight gain and showing a less-reactive intestine and liver condition. These nutraceutical effects would improve the ability of fish to deal with production conditions or with modifications in the diet (use of different ingredients). Here, for the first time, we demonstrate the direct effects of dietary LPLs on the proteinteractome of a marine cultured species, opening a new window into the study of the benefits of LPL supplementation in fish nutrition that could be extended to other productive species.

## Figures and Tables

**Figure 1 animals-13-01381-f001:**
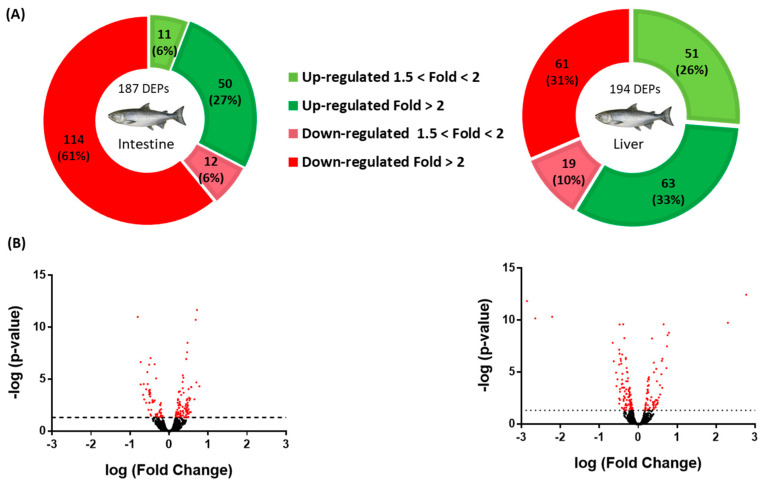
Analyses of differentially expressed proteins (DEPs) in the intestine and liver of salmon fed a lysophospholipid-enriched diet. (**A**) Pie charts for the intestine and liver DEPs. The green (upregulation) and red (downregulation) color scheme indicates the protein modulation according to its fold-change magnitude interval. (**B**) Volcano plots for the intestine and liver DEPs. The Volcano plots were constructed by plotting the negative logarithm of the *p*-value on the y-axis.

**Figure 2 animals-13-01381-f002:**
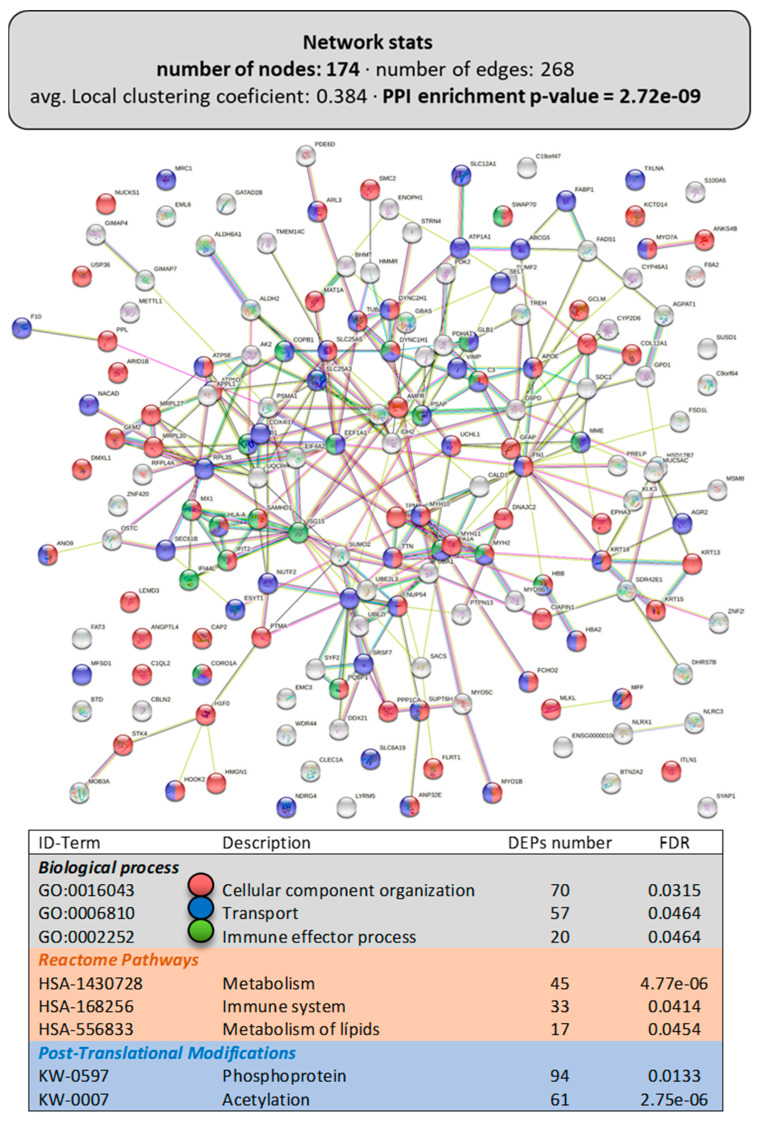
Functional enrichments in the interactome of the intestine of LPL-fed salmon. Each node represents one differentially expressed protein (DEP) obtained from mucus shotgun proteomics (details in Appendix A). Details on the functional network statistics (network stats) are indicated (**top**). The selected relevant biological processes (**bottom**) are presented with the corresponding colors in each node. The main Reactome pathways and post-translational modifications (PTMs), clustered significantly by the false discovery rate (FDR) value, are also summarized (**bottom table**).

**Figure 3 animals-13-01381-f003:**
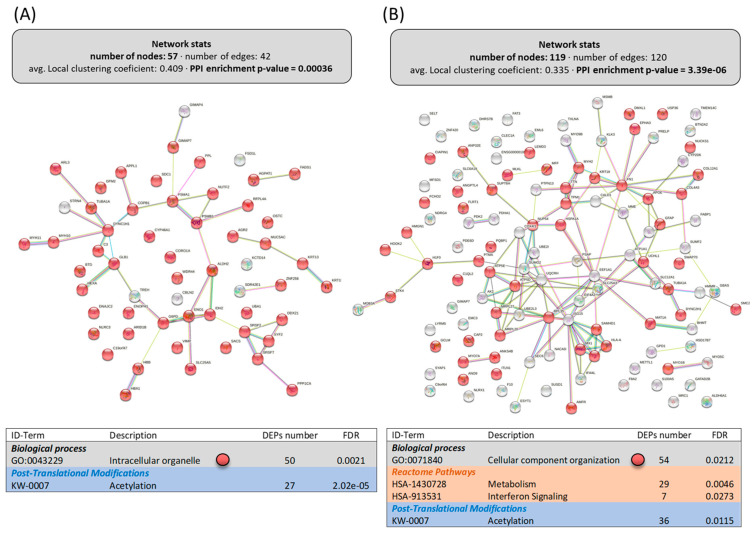
Functional enrichments in the interactomes of upregulated (**A**) and downregulated (**B**) DEPs in the intestine of LPL-fed salmon. Each node represents one differentially expressed protein (DEP) obtained from mucus shotgun proteomics. Details on the functional network statistics (network stats) are indicated (**top**). The selected relevant biological processes (**bottom**) are presented with the corresponding colors in each node. The main Reactome pathways and post-translational modifications (PTMs), clustered significantly by the false discovery rate (FDR) value, are also summarized (**bottom table**).

**Figure 4 animals-13-01381-f004:**
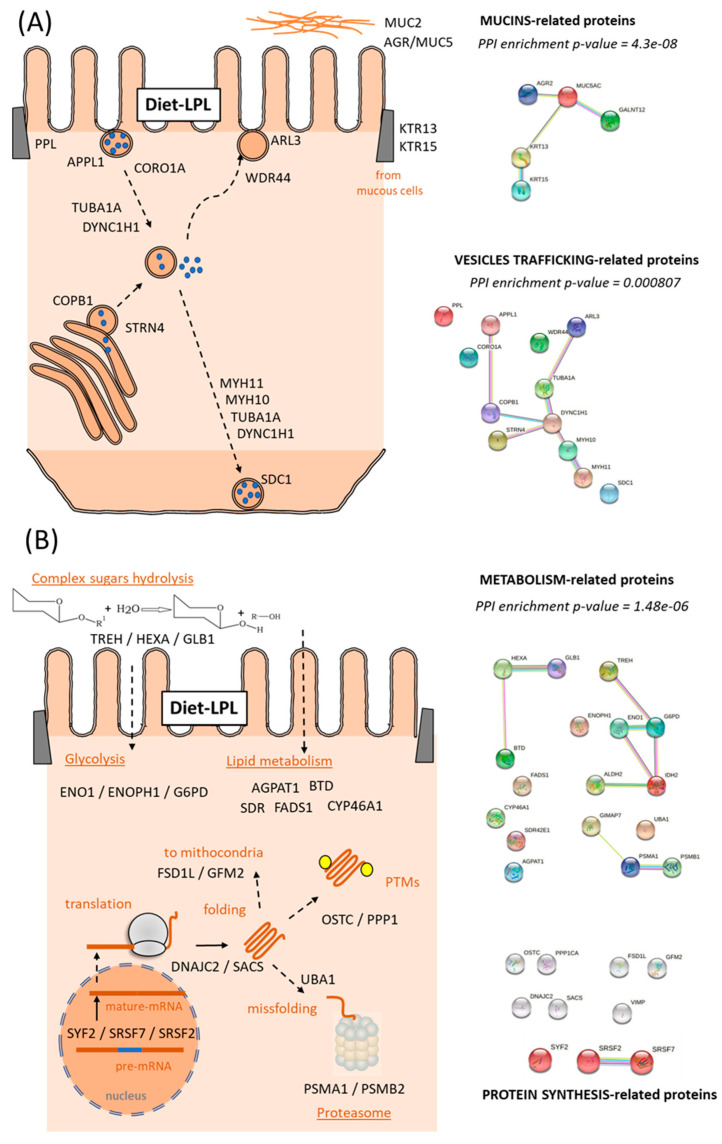
Diagrams of the functional pathways upregulated by the LPL-diet in the salmon intestine. (**A**) Vesicles Trafficking and Extracellular Matrix and (**B**) Metabolic-related proteins. Putative relationships of the upregulated DEPs from their main function (UniProtKB). For each cluster, the protein–protein interaction (PPI)-enrichment *p*-values are shown. Details of each DEP are described in the Appendix A.

**Figure 5 animals-13-01381-f005:**
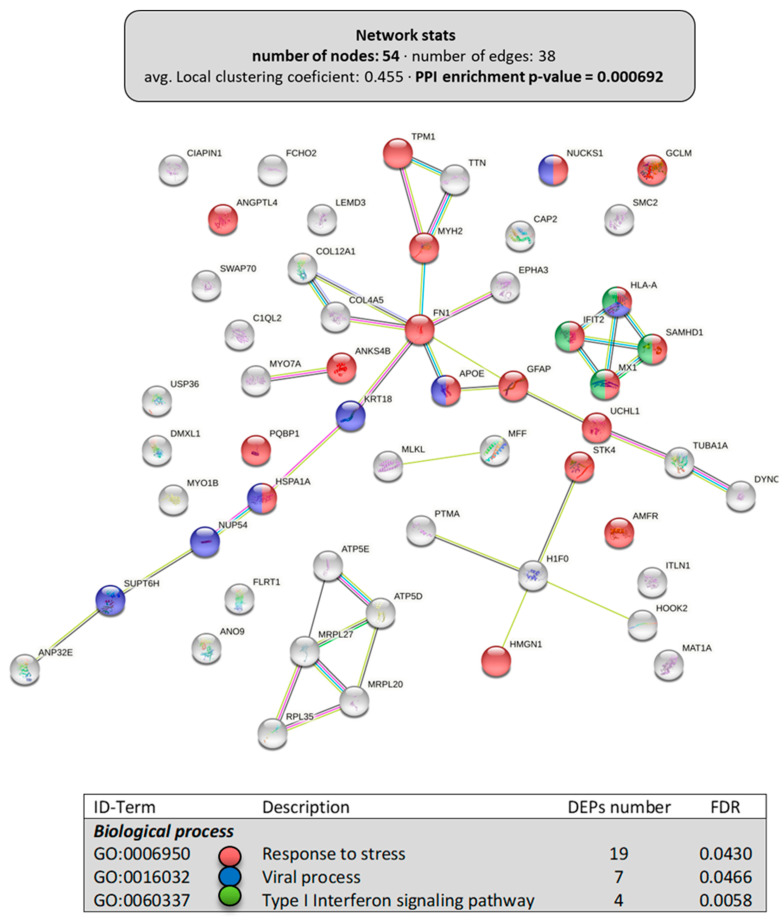
Specific network of the downregulated DEPs in the intestine of LPL-fed salmon. Analyses of putative clusters related to “cellular component organization” of the downregulated DEPs (Figure 3). Each node represents one differentially expressed protein (DEP) obtained from intestine shotgun proteomics. Details on the functional network statistics (network stats) are indicated (**top**). The selected relevant biological processes (**bottom**) are presented with the corresponding colors in each node.

**Figure 6 animals-13-01381-f006:**
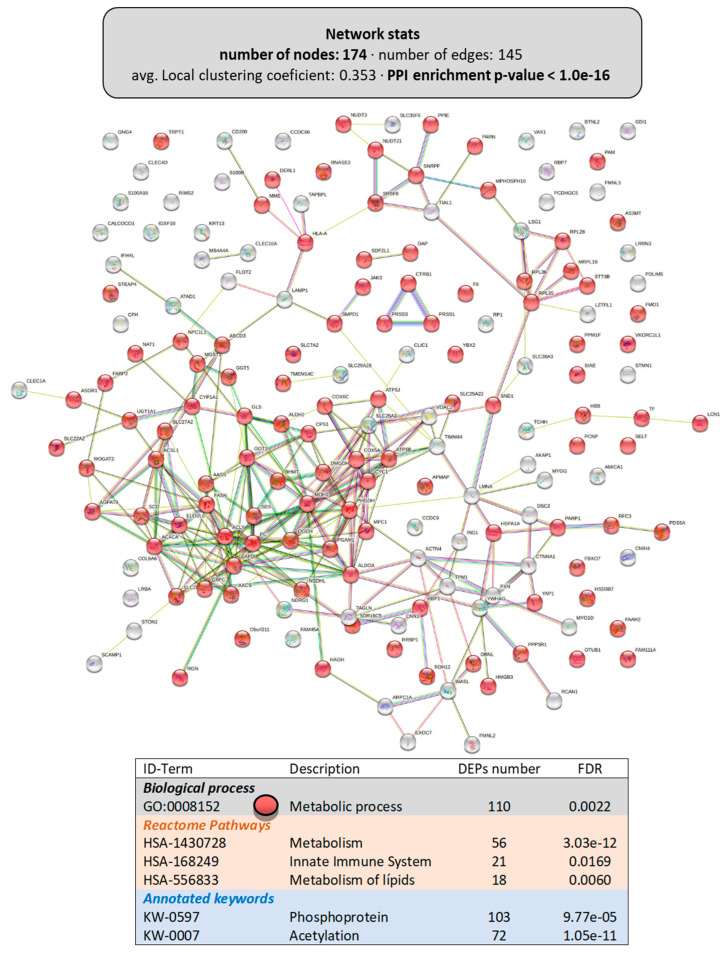
Functional enrichments in the interactome of the liver of LPL-fed salmon. Each node represents one differentially expressed protein (DEP) obtained from liver shotgun proteomics (details in Appendix A). Details on the functional network statistics (network stats) are indicated (**top**). The selected relevant biological processes (**bottom**) are presented with the corresponding colors in each node. The main Reactome pathways and post-translational modifications (PTMs), clustered significantly by the false discovery rate (FDR) value, are also summarized (**bottom table**).

**Figure 7 animals-13-01381-f007:**
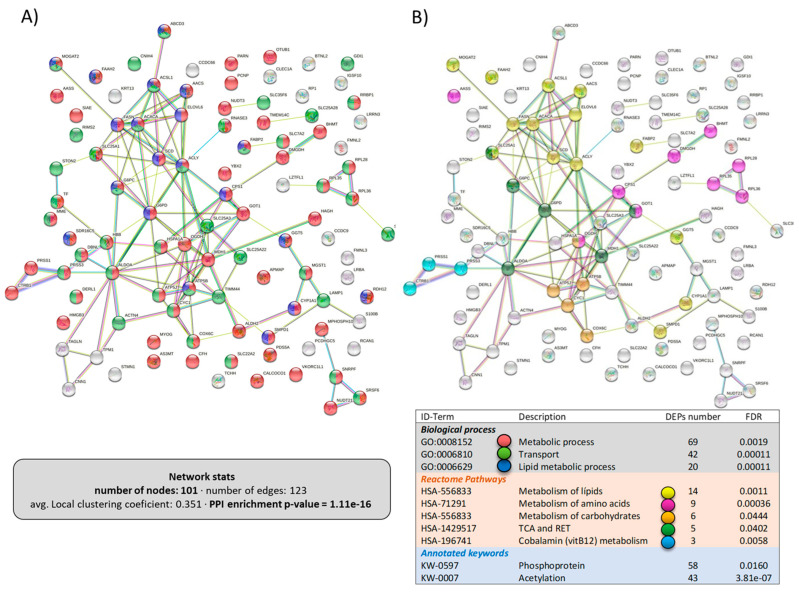
Specific network of the upregulated DEPs in the liver of LPL-fed salmon. Each node represents one differentially expressed protein (DEP) obtained from liver shotgun proteomics. Details on the functional network statistics (network stats) are indicated. The selected clusters are presented with the corresponding colors in each node for (**A**) the biological process and (**B**) the Reactome pathways.

**Figure 8 animals-13-01381-f008:**
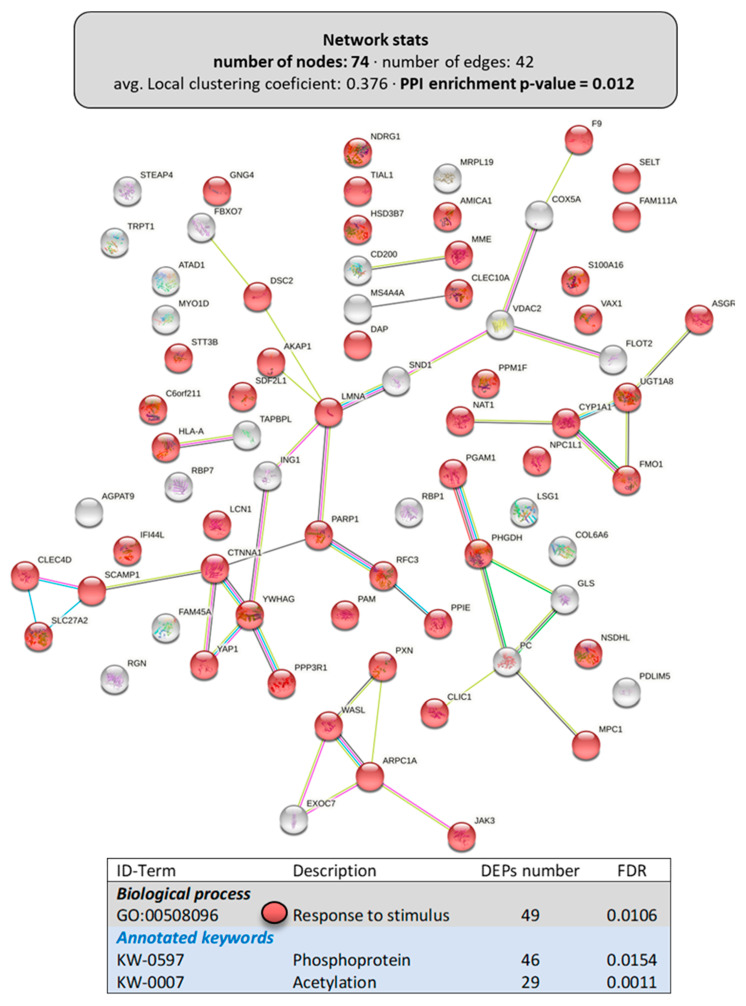
Specific network of the downregulated DEPs in the liver of LPL-fed salmon. Each node represents one differentially expressed protein (DEP) obtained from liver shotgun proteomics. Details on the functional network statistics (network stats) are indicated (**top**). The selected relevant biological processes are presented with the corresponding colors in each node, together with the annotated keywords related to post-translational modifications (**bottom table**).

**Figure 9 animals-13-01381-f009:**
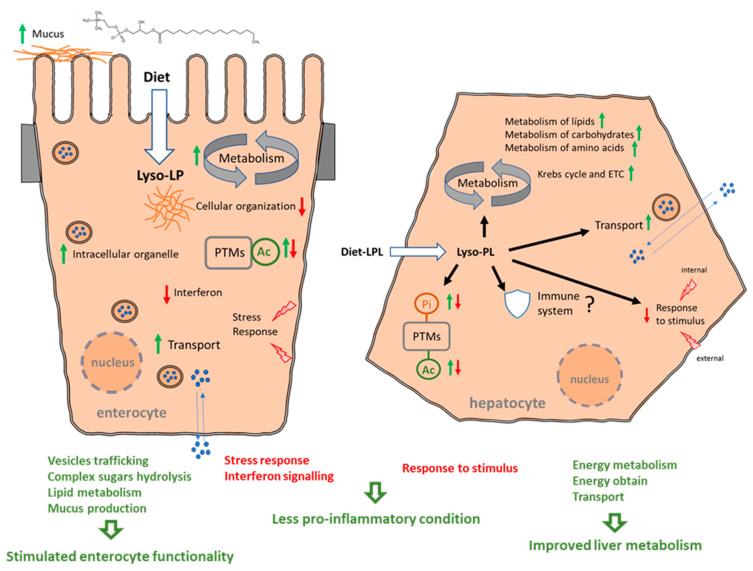
Summary of the modes of action of dietary LPLs in salmon intestine and liver. Green arrows mean upregulated processes and red arrows indicated downregulated processes.

**Table 1 animals-13-01381-t001:** Growth performance of Atlantic salmon fed with control diet (C-diet) and lysophospholipid-enriched diet (LPL-diet).

	C-Diet	LPL-Diet	*p*-Value (Student’s *t*-Test ^1^)
Final weight (g)	477 ± 6	501 ± 6	0.009
Final length (cm)	32.0 ± 0.1	32.3 ± 0.1	-
Condition Factor ^2^ (%)	1.45 ± 0.01	1.48 ± 0.01	0.017
SGR ^3^	1.22 ± 0.09	1.29 ± 0.01	-
TGC ^3^	2.48 ± 0.22	2.65 ± 0.06	-
FCR ^3^	0.74 ± 0.02	0.74 ± 0.02	-
HIS ^4^	0.88 ± 0.03	0.99 ± 0.03	0.015

Data expressed as mean ± SEM (*n* =105 per condition from 3 triplicate tanks). Each replicate represented by an individual fish. ^1^ Student *t*-test analysis only reported when *p*-value < 0.05; ^2^ condition factor; ^3^ specific growth rate (SGR), thermal growth coefficient (TGC), and feed conversion rate (FCR) were calculated by *n* = 3 replicate tanks, and corresponding formulas and units are specified in Material and Methods section; ^4^ hepato-somatic index (*n* = 10 sampled fish).

**Table 2 animals-13-01381-t002:** Plasma metabolites of Atlantic salmon fed with control diet (C-diet) and lysophospholipid-enriched diet (LPL-diet).

	C-Diet	LPL-Diet	*p*-Value (Student’s *t*-Test ^1^)
Protein (mg/mL)	51.0 ± 0.7	51.1 ± 0.3	-
Glucose (mg/dL)	121 ± 6	110 ± 5	-
Triglycerides (mg/dL)	306 ± 21	292 ± 15	-
Phospholipids (mg/dL)	736 ± 35	666 ± 15	0.032
Total cholesterol	689 ± 33	634 ± 21	-
Sum of lipid fractions ^2^ (mg/dL)	1731 ± 11	1467 ± 68	0.041
Lactate-DH ^3^ (U/mL)	2820 ± 255	1950 ± 125	0.023

Data expressed as mean ± SEM (*n* = 10). Each replicate represented by an individual fish. ^1^ Student *t*-test analysis only reported when *p*-value < 0.05; ^2^ sum of individual values of each lipid fraction in plasma; ^3^ activity of plasma lactate dehydrogenase.

## Data Availability

Not applicable.

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
