# Peer review of "Physiological Benefits of Dietary Lysophospholipid Supplementation in a Marine Fish Model: Deep Analyses of Modes of Action"

_animals, 2023, doi:10.3390/ani13081381_

Round 1
Reviewer 1 Report
1.L87 “The experimental feeds were then administered to four two triplicate groups of Atlantic salmon in a 12-week experiment.”
The description for groups is unclearly. It's four groups? two groups? How many replicates were there in each group?
2.L83-84 Please normalize the units of area and volumeof the tanks.
3.L92-93"An overfeeding by 20% was undertaken to allow maximum feed intake."
How did you tell if the fish was full?
4.L91-92 The author said the fish ingested one meal every 3 h and 8 meals per day through automatic disc feeders.
L104 The authors said fish were fed again,however,the animals were fed two times a day for 12 weeks.
What is the feeding strategy of the experiment?
5.L106 What was the overdose? Please provide the accurate dose used in the trial.
6."2.1 feeding trail" included too many informance, such as feed prepared, feeding manegment, fish,ect. Pleae made these informace logic.
7. 2.1 part:The formula information was missing here. How did the author prepare the feed and how did lysophospholipid add to the feed?
8.Table 1,Please used condition factor replaced the "k",which is irregular. Please provide the calculation formula for SGR,K,THC,FCR, and HSI.
Please explain the reason of n=105.
9.the authors emphasized that digestibility enhancers aimed to maximize lipid absorption and utilization, and were categorized as emulsifiers. however, they didn't test the digestive enzymes.
10.discussion
The authors did not make sufficient use of the experimental data to discuss the changes in growth.
Author Response
Reviewer 1:
1.L87 “The experimental feeds were then administered to four two triplicate groups of Atlantic salmon in a 12-week experiment.”
The description for groups is unclearly. It's four groups? two groups? How many replicates were there in each group?
This sentence has been modified as follows:
“The experimental feeds were then administered to two triplicate groups of Atlantic salmon in a 12-week experiment.”
2.L83-84 Please normalize the units of area and volume of the tanks.
The units have been normalized.
3.L92-93"An overfeeding by 20% was undertaken to allow maximum feed intake."
How did you tell if the fish was full?
4.L91-92 The author said the fish ingested one meal every 3 h and 8 meals per day through automatic disc feeders.
Initial fish weight was around 160g. Following the recommendations of commercial feeds for this size of salmon, diet was adjusted as described (8 times per day, every 3h) with a daily estimated overdose of 20% to ensure the achievement of daily dose. Weekly, the dose was readjusted assuming an FCR of 0.8.
As we described later in the M&M: “All the feed spill was collected from the water outlet and the feed intake was quantified as the difference between the delivered and wasted feed, after correcting for dry matter content and feed spill recovery”
L104 The authors said fish were fed again, however,the animals were fed two times a day for 12 weeks.
What is the feeding strategy of the experiment?
This sentence has been removed. We apologies the mistake.
5.L106 What was the overdose? Please provide the accurate dose used in the trial.
The anaesthetic doses recommended for Atlantic salmon and rainbow trout, 50–60 mg L-1, whereas the doses over 400 mg/L is generally used for euthanasia of salmonids.
6."2.1 feeding trail" included too many informance, such as feed prepared, feeding manegment, fish,ect. Pleae made these informace logic.
With the aim to clarify the reading of this chapter, the text has been restructured in 4 different paragraphs, including the before-mentioned concerns (see revised version of the manuscript).
- 2.1 part:The formula information was missing here. How did the author prepare the feed and how did lysophospholipid add to the feed?
All diets were based on a common basal formulation for salmon with 20 % fish meal, developed by MOWI (confidential). The feed manufacturing process was done in Nofima as a pilot scale version of the process used industrially for production of salmonid feeds, and involves mixing of dry ingredients, pelleting by extrusion, and coating with the residual oil. The additive was added to the mixer before extrusion replacing 0.1% of the rapeseed oil.
The following sentence has been added in the Material and Methods section:
“The diets was based on a common basal formulation for salmon with 20% of fish meal and the LPLs were added to the mixer before extrusion replacing 0.1% of the rapeseed oil.”
8.Table 1,Please used condition factor replaced the "k",which is irregular. Please provide the calculation formula for SGR,K,THC,FCR, and HSI.
The formulas have been added to the M&M section.
Please explain the reason of n=105.
The following sentence has been added in the Table legend: (n =105 per condition from 3 triplicate tanks).
9.the authors emphasized that digestibility enhancers aimed to maximize lipid absorption and utilization, and were categorized as emulsifiers. however, they didn't test the digestive enzymes.
We agree with the Referee in the relevance of the digestive enzyme’s activities, once we have demonstrated the benefits on enterocytes. When we planned the study, we considered that many studies have demonstrated the beneficial effects of dietary phospholipids on fat digestibility in fish (see Introduction references), however, the physiological mechanisms underlying the modes of action or additional metabolic effects of LPLs have not been explored yet.
Now, we have added the following sentence in the Discussion chapter: “Regarding these findings on enterocyte functionality, in further studies should also approach the role of the digestive enzymes’ activities related lipid digestion the enterocyte absorption in salmon, and other fish or shrimp species, fed with LPLs”.
10.discussion
The authors did not make sufficient use of the experimental data to discuss the changes in growth.
This paragraph of the discussion has been revised according the comments from all referees in the revised version of the manuscript.
Reviewer 2 Report
The manuscript presents original data on the effects of LPLs on growth performance, lipid utilization, enterocyte function, and liver metabolism. It suggests some new avenues in applying LPLs in fish feeds. The experimental design and data analysis are generally sound. However, the study discussion and conclusions are speculative at times, and need some re-phrasing.
Suggested changes;
Lines 82–83 – Correct Unit of surface and volume with superscripts
Line 88 – Please correct sentence “administered to four two triplicate groups of Atlantic salmon…”.
Line 412 and further in the same paragraph– The wording “A numerical (5% higher) increase..”, and similar, may be misleading. If one says “higher” it should be backed by statistical significance, otherwise you may use “ SGR shows also a tendency for higher values”.
Line 428-430 - The sentence “in response to LPL supplementation would indicate that the fish were able to consume higher amounts of lipids when compared to control conditions.” Should be re-rewritten according to the feed intake data (see remark about table 1).
Line 476 – Where it stands “protein synthesis functions”, I suggest to use “protein synthesis and turnover functions”, as some of regulated proteins, e.g., the ones involved ion proteasomal activity, are clearly not involved in protein synthesis sensu stricto, but rather in protein turnover. Even if the Go term is “protein synthesis functions”, I think and adaption would make it more clear.
Lines 537-539 – The authors propose that the observed “less-reactive condition” of fish receiving LPL supplementation, suggested by the interactome analysis, may be attributed to a general reduced proinflammatory state in LPL-fed fish. They should justify this indicating some examples of DEP involved in proinflammatory response.
Lines 545-546 – The authors propose a new hypothesis; an additional mode of action of LPLs is in the modulation of protein function. However, the evidence provided seems circumstantial, as DEPs detected being putative targets of PTMs does not mean their expression is being regulated by PTMs. Also the discussion in the following lines does not really provide evidence on this. I suggest to remove or tone-down this paragraph.
Line 553-554 – Please provide references.
Line 558 – The claim that LPLs reduced the proinflammatory condition in the present study, seems excessive. I suggest to remove it. Even if some of the DEPs may be involved in inflammatory response, no evidence is provided on the inflammatory condition. It is fine to hypothesize in the discussion, but having it in the conclusions is a big leap. The “Simple Summary” should also be toned-down in this respect.
Line 560-564 – The final sentence seems wordy and adds little to the actual conclusions of the study. I suggest to remove it.
Table 1. SGR normally stands for “specific growth rate”, not “Standard growth rate”. Please clarify what did you use, and the formula for the calculation and units . Also if this is specific growth rate, the values given are most likely mulipiled by 100, which would give a %/day unit, what is wrong. The right unit should be /day (day-1), see Halver and Hardy textbook. Also FCR normally stands for Feed (not Food) conversion rate. Moreover in this table feed intake is not given, even if it was measured according to M&M. Please provide these data.
Author Response
Reviewer 2:
The manuscript presents original data on the effects of LPLs on growth performance, lipid utilization, enterocyte function, and liver metabolism. It suggests some new avenues in applying LPLs in fish feeds. The experimental design and data analysis are generally sound. However, the study discussion and conclusions are speculative at times, and need some re-phrasing.
Suggested changes;
Lines 82–83 – Correct Unit of surface and volume with superscripts
The units have been normalized.
Line 88 – Please correct sentence “administered to four two triplicate groups of Atlantic salmon…”.
This sentence has been modified as follows:
“The experimental feeds were then administered to two triplicate groups of Atlantic salmon in a 12-week experiment.”
Line 412 and further in the same paragraph– The wording “A numerical (5% higher) increase..”, and similar, may be misleading. If one says “higher” it should be backed by statistical significance, otherwise you may use “ SGR shows also a tendency for higher values”.
In agreement with the referee comment, we have modified the sentence as follows:
“SGR also showed a tendency for higher values, whereas FCR did not change.”
Line 428-430 - The sentence “in response to LPL supplementation would indicate that the fish were able to consume higher amounts of lipids when compared to control conditions.” Should be re-rewritten according to the feed intake data (see remark about table 1).
To avoid confusions, we have removed this sentence of the Discussion chapter.
Line 476 – Where it stands “protein synthesis functions”, I suggest to use “protein synthesis and turnover functions”, as some of regulated proteins, e.g., the ones involved ion proteasomal activity, are clearly not involved in protein synthesis sensu stricto, but rather in protein turnover. Even if the Go term is “protein synthesis functions”, I think and adaption would make it more clear.
The term “turnover” has bee added to this sentence.
Lines 537-539 – The authors propose that the observed “less-reactive condition” of fish receiving LPL supplementation, suggested by the interactome analysis, may be attributed to a general reduced proinflammatory state in LPL-fed fish. They should justify this indicating some examples of DEP involved in proinflammatory response.
As we indicated in the Results and Discussion sections, there were 19 DEPs in intestine significantly down-regulated and grouped in the GO “response to stress” and 49 DEPs in liver down-regulated and grouped in the GO “response to stimulus”. All these DEPs are highlighted in the Figures 5 and 8, respectively, and the corresponding data from each DEP are included in the Supplementary Files. Due to the great number of the DEPs we considered to not focus on specific DEP but on this “less-reactive condition”.
However, in agreement to this referee comment, we modified the sentence including specific Biological Process in intestine also significant such as “viral process” and “interferon signalling pathway.
“Regarding the downregulated DEPs in the intestine and liver of LPL-fed fish, they clustered mainly in biological processes related to tissue reactivity, such as the “re-sponse to stress” (GO:0006950, with 19 DEPs), which included 11 DEPs significantly grouped as “viral process” (7 DEPs) and “interferon signaling pathway” (4 DEPs) in the intestine, and the “response to stimulus” (including 49 of the 74 downregulated DEPs) in the liver.
Lines 545-546 – The authors propose a new hypothesis; an additional mode of action of LPLs is in the modulation of protein function. However, the evidence provided seems circumstantial, as DEPs detected being putative targets of PTMs does not mean their expression is being regulated by PTMs. Also the discussion in the following lines does not really provide evidence on this. I suggest to remove or tone-down this paragraph.
In agreement with this comment, we have modified the paragraph with the aim to tone-down the relevance of the LPLs on PTMs pathways.
“Finally, the study of changes in the proteinteractome caused by dietary LPLs revealed that most of the DEPs detected were putative targets of post-translational modification (PTM), mainly acetylation and phosphorylation. PTMs are reversible processes that can dynamically regulate the metabolic state of cells through regulating protein structure, activity, localization or protein–protein interactions. PTMs can activate or inactivate catalytic functions or influence the biological activity of a protein [55]. There not exists specific information on the direct relationship between LPLs and PTMs, but some evidence indicates a clear role of LPLs in cellular signaling. Diverse effects of LPLs on cell responses include mitogenesis, differentiation, cell migration, and cell viability (anti-apoptosis) (reviewed in [56]). As reversible protein phosphorylation and acetylation are major modifications regulating protein function and lipid second messengers can modulate these cellular signaling pathways [56], the role of the dietary LPLs needs to be investigated and further and deeper studies are necessary in this field.
Line 553-554 – Please provide references.
This sentence has been removed in the revised manuscript.
Line 558 – The claim that LPLs reduced the proinflammatory condition in the present study, seems excessive. I suggest to remove it. Even if some of the DEPs may be involved in inflammatory response, no evidence is provided on the inflammatory condition. It is fine to hypothesize in the discussion, but having it in the conclusions is a big leap. The “Simple Summary” should also be toned-down in this respect.
The sentence “reducing the proinflammatory condition” has been replaced by “showing a less-reactive intestine and liver condition” in the conclusion and in the simple summary sections.
Line 560-564 – The final sentence seems wordy and adds little to the actual conclusions of the study. I suggest to remove it.
The last sentence has been modified as follows: “opening a new window into the study of the benefits of LPL supplementation in fish nutrition and could be extended to other productive species.
Table 1. SGR normally stands for “specific growth rate”, not “Standard growth rate”. Please clarify what did you use, and the formula for the calculation and units . Also if this is specific growth rate, the values given are most likely mulipiled by 100, which would give a %/day unit, what is wrong. The right unit should be /day (day-1), see Halver and Hardy textbook. Also FCR normally stands for Feed (not Food) conversion rate. Moreover in this table feed intake is not given, even if it was measured according to M&M. Please provide these data.
Accordingly, the terms “standard growth rate” has been replaced by the term “specific growth rate”, and the term “food conversion rate” by “feed conversion rate”. We regret these mistakes.
All formula included in Table 1 have been added in the Material and Methods
Reviewer 3 Report
Review for the paper "Physiological Benefits of Dietary Lysophospholipid Supplementation in a Marine Fish Model: Deep Analyses of Modes of Action" by Antoni Ibarz, Ignasi Sanahuja, Waldo G. Nuez-Ortín, Laura Martínez-Rubio, and Laura Fernández-Alacid submitted to "Animals".
General comment.
Aquaculture has been the world’s fastest growing food producing industry during the past decades, with Atlantic salmon being among the most successful aquaculture species in production growth resulting from improved inputs, better production practices at the farms, improved logistics and more efficient supply chains. The development of new aquafeeds and ingredients is the most promising way in intensifying this industry and for successful practices, baseline data on the mechanisms underlying the metabolic processes are required. However, there is a lack of information regarding the putative mode of action of lysophospholipids in salmon nutrition. The authors conducted a pilot study to reveal the effects of an LPL-based digestive enhancer on the growth performance and serum indices in Atlantic salmon. This paper is well-written but some important issues should be addressed. The authors observed an increase in body weight and condition factor and declared positive effects, but an increased HSI and liver weight contradict this opinion.
Major concerns.
The authors stated that “The LPL-diet had a positive effect on the fish by increasing the final weight by 5%”. However, this effect seems to be not quite favorable because it is explained by an increase in the liver size, increase in phosphorlipid content and, although insignificant, an increase in HSI and total cholesterol, while other important parameters, especially protein content, were not affected. Moreover, this diet modified “metabolism of lipids“ (L 275) and caused “the response to stress in the intestine” (L 375). It is known that an increased HSI is an indicator of unfavorable conditions.
L 383-392. The authors summarized positive effects according to the analyses of the intestinal and liver interactomes but it is unclear why these effects are considered positive. These may be a reaction to stress. A longer rearing period is needed to clearly conclude if the effect is positive or negative.
As I know, salmon aquaculture is focused on flesh, not on liver, but the authors provided no information on flesh quality and applicability of this diet to obtain a high-quality product.
The authors described the possible mechanisms but the rearing period is too short to obtain reliable information because a cycle of entire salmon farming lasts for several years.
Abstract
The authors described the positive effect on growth performance, but details about physiological parameters were not present. The authors should update the Abstract.
Introduction
The authors should provide a more detailed description of their study object, Atlantic salmon, including trends in its global aquaculture production and prices.
L 68. Consider replacing “the perfect candidate” with “a perfect candidate”
Material and methods.
L 88. It is unclear what do mean “four two triplicate groups”
L 194. Please, define the abbreviation “FDR” (false discovery rate).
L 202. “Homo sapiens” should be italicized
L 204. “H. sapiens” should be italicized
L 216. Prior to this parametric analysis, the data must be checked for normality and homogeneity of variance. Please, update the text accordingly.
The authors should provide the formulas for growth performance indices.
Results
L 229 “after a 3-month” should be “after the 3-month”
The text in Figure 1 is not easy to read due to the small size and an unfortunate choice of colors. Please, re-draw.
In the current forms, Figures 2, 5, 6, 7 provide interesting pictures but of low significance, because the font size for core information is too small.
In Fig. 8, it is also small but the reader, with some effort, can obtain the necessary information.
Fig. 4 contains a number of unencrypted abbreviations, which prevent clear understanding of the processes.
Discussion.
L 395-403. The authors should provide relevant citations
L 421. Consider replacing “has been observed on” with “has been observed in”
L 412. The authors suggested an increase in SGR, but, according to Table 2, this increase was not significant
-Discuss the limitations of the study.
Author Response
Reviewer 3
General comment.
Aquaculture has been the world’s fastest growing food producing industry during the past decades, with Atlantic salmon being among the most successful aquaculture species in production growth resulting from improved inputs, better production practices at the farms, improved logistics and more efficient supply chains. The development of new aquafeeds and ingredients is the most promising way in intensifying this industry and for successful practices, baseline data on the mechanisms underlying the metabolic processes are required. However, there is a lack of information regarding the putative mode of action of lysophospholipids in salmon nutrition. The authors conducted a pilot study to reveal the effects of an LPL-based digestive enhancer on the growth performance and serum indices in Atlantic salmon. This paper is well-written but some important issues should be addressed. The authors observed an increase in body weight and condition factor and declared positive effects, but an increased HSI and liver weight contradict this opinion.
Major concerns.
The authors stated that “The LPL-diet had a positive effect on the fish by increasing the final weight by 5%”. However, this effect seems to be not quite favorable because it is explained by an increase in the liver size, increase in phosphorlipid content and, although insignificant, an increase in HSI and total cholesterol, while other important parameters, especially protein content, were not affected. Moreover, this diet modified “metabolism of lipids“ (L 275) and caused “the response to stress in the intestine” (L 375). It is known that an increased HSI is an indicator of unfavorable conditions.
We agree with the referees in the consideration that an increase on liver size could be classicaly related with a worst animal condition. However, the HSI values are very low in both control and LPL-diets (0.88 vs 0.99, respectively). Even, we consider that there is a better optimization of the use of nutrients, it is possible to see a healthier liver, more developed than control.
“increase in phosphorlipid content and, although insignificant, an increase in HSI and total cholesterol,”
Both increments in phospholipids and total cholesterol are not reported in liver. Instead, we did not analyse these lipid fractions in liver. On the contrary the levels in plasma were reduced, and it is explained in discussion we consider these levels as indicative of a better condition.
“this diet modified “metabolism of lipids“ (L 275)”
The DEPs included in the up-regulated GO “metabolism of lipids” in liver demonstrated that the metabolization of lipids in liver are improved by the dietary LPLs, as well as, other metabolic processes as carbohydrates and protein, which implies a better liver condition.
“the response to stress in the intestine”
This GO was down-regulated in intestine, indicating from our point of view a less-reactive intestine of animals fed LPLs versus fed diet without LPLs
L 383-392. The authors summarized positive effects according to the analyses of the intestinal and liver interactomes but it is unclear why these effects are considered positive. These may be a reaction to stress. A longer rearing period is needed to clearly conclude if the effect is positive or negative.
We agree with the referee concern on the short rearing period used here, which is the main limitation of the current study. However, as we tried to explain in Discussion chapter, the up-regulation, for instance, of vesicles trafficking in intestine or the up-regulated metabolism in liver should be considered as positive effects in these tissues, such those are one of their main functions. In parallel, both tissues showed reduced reactivity, with downregulations in the stress response (intestine) or in the response to stimuli (liver).
We would like indicated here that in response to other concerns from all referees we “tone-down” the expressions related to “a reduction in proinflammatory state” being them removed in the revised manuscript.
As I know, salmon aquaculture is focused on flesh, not on liver, but the authors provided no information on flesh quality and applicability of this diet to obtain a high-quality product.
We absolutely agree with this concern. However, this is first study did not include the effects on muscle quality. We are very interested in this future study as the dietary LPLs should also improved the phospholipid intake and this is a current focus for industry as we have indicated in Introduction: “salmon flesh coloration, which is a key commercial trait of wild and farmed salmon alike [15], can be improved by the lipid-associated absorption of carotenoids [16]”. In agreement, we have introduced the following sentence in the Discussion chapter:
“In the view of current results in intestine and liver functionality, the study of flesh quality and the applicability of this diet to obtain a high-quality product could be a next step with a major interest of salmon industry.”
The authors described the possible mechanisms but the rearing period is too short to obtain reliable information because a cycle of entire salmon farming lasts for several years.
We absolutely agree with this concern. We consider the current study as a pilot study with the aim to determine the effects and mode of action of dietary LPLs in salmon. This critical limitation of the study has been included in the Discussion chapter and Conclusion.
Abstract
The authors described the positive effect on growth performance, but details about physiological parameters were not present. The authors should update the Abstract.
Although we agree with the referee, the Instructions for authors of the Animals journals indicated that “The abstract should be a total of about 200 words maximum”. This instruction limits the length of the abstract including few informative results.
Introduction
The authors should provide a more detailed description of their study object, Atlantic salmon, including trends in its global aquaculture production and prices.
We have introduced the following sentences in the Introduction:
“The Atlantic salmon represents 32.5% of global marine finfish aquaculture production and is the fourth most economically important farmed species. Current worldwide production of farmed Atlantic salmon exceeds 1 000 000 tonnes, where farmed Atlan-tic salmon constitute over the 90% of the farmed salmon market, and over the 50% of the total global salmon market. (FAO, 2022; https://www.fao.org).”
L 68. Consider replacing “the perfect candidate” with “a perfect candidate”
Done
Material and methods.
L 88. It is unclear what do mean “four two triplicate groups”
This sentence has been corrected in the revised manuscript.
L 194. Please, define the abbreviation “FDR” (false discovery rate).
The false discovery rate has been previously defined 7 lines before the mention in L194 (of the original manuscript)
L 202. “Homo sapiens” should be italicized
Done
L 204. “H. sapiens” should be italicized
Done
L 216. Prior to this parametric analysis, the data must be checked for normality and homogeneity of variance. Please, update the text accordingly.
The following sentence has been added: “The data were checked for normality and homoscedasticity prior to analysis.”
The authors should provide the formulas for growth performance indices.
All formula included in Table 1 have been added in the Material and Methods. We regret this mistake.
Results
L 229 “after a 3-month” should be “after the 3-month”
Done
The text in Figure 1 is not easy to read due to the small size and an unfortunate choice of colors. Please, re-draw.
The Figure 1 has been modified with these suggestions in the revised manuscript.
In the current forms, Figures 2, 5, 6, 7 provide interesting pictures but of low significance, because the font size for core information is too small.
These Figures have been revised, improved and slightly enlarged when necessary.
In Fig. 8, it is also small but the reader, with some effort, can obtain the necessary information.
The Figure 8 has been improved.
Fig. 4 contains a number of unencrypted abbreviations, which prevent clear understanding of the processes.
We have added the following text to clarify: “Details of each DEPs are described in the Supplementary File 5”.
Discussion.
L 395-403. The authors should provide relevant citations
Some references before-mentioned in Introduction have been added.
L 421. Consider replacing “has been observed on” with “has been observed in”
L 412. The authors suggested an increase in SGR, but, according to Table 2, this increase was not significant
In agreement with the comments from the referees 1, 2 and 3, these sentences have been revised.
-Discuss the limitations of the study.
In response to major concerns highlighted for all referees, the limitations of the study have been included in the revised manuscript at different points of the discussion.
Round 2
Reviewer 1 Report
1. why did the Final weight has the unit and the other index didn't in Table 1?
2.L180 Was the format of the data wrong?
Author Response
Dear Referee, Thanks agains for your revision.
- why did the Final weight has the unit and the other index didn't in Table 1?
To clarify, we have added for Condition Factor the symbol (%) and the following sentence in Table text: "Corresponding formulas and units are specified in Material and Methods section".
2.L180 Was the format of the data wrong?
We do not know exactly what the referee refers to with this indication. The data indicated in the L180 are correct according to the information obtained from the mass spectrometry.
Reviewer 3 Report
In general, the authors have considered my recommendations.
Author Response
Thanks for the second revision of our manuscript.
Nos gustaría indicar que el manuscrito original ha sido revisado por un corrector nativo inglés de los Servicios Lingüísticos de la Universidad de Barcelona.
Las oraciones añadidas en la versión revisada ahora se verifican y se han detectado y revisado algunos errores.
For instance: L99 "was" by "were".
Sentences in L481-485: rewritten.